



# Subseasonal prediction of springtime Pacific-North American transport using upper-level wind forecasts

John R. Albers[1,2], Amy H. Butler[3], Melissa L. Breeden[3], Andrew O. Langford[3], George N. Kiladis[2]

[1]Cooperative Institute for Research in the Environmental Sciences, University of Colorado Boulder, Boulder, 80305, USA
[2]Physical Science Laboratory, NOAA Earth System Research Laboratory, Boulder, 80305, USA
[3]Chemical Science Laboratory, NOAA Earth System Research Laboratory, Boulder, 80305, USA

*Correspondence to*: John R. Albers (john.albers@noaa.gov)

**Abstract.** Forecasts of Pacific jet variability are used to predict stratosphere-to-troposphere transport (STT) and tropical-to-extratropical moisture exports (TME) during boreal spring over the Pacific-North American region. A retrospective analysis
first documents the regionality of STT and TME for different Pacific jet patterns. Using these results as a guide, Pacific jet hindcasts, based on zonal-wind forecasts from the European Centre for Medium-Range Weather Forecasting Integrated Forecasting System, are utilized to test whether STT and TME over specific geographic regions may be predictable for subseasonal forecast leads (3-6 weeks ahead of time). Large anomalies in STT to the mid-troposphere over the North Pacific, TME to the west coast of the United States, and TME over Japan are found to have the best potential for subseasonal
predictability using upper-level wind forecasts. STT to the planetary boundary layer over the intermountain west of the United States is also potentially predictable for subseasonal leads, but likely only in the context of shifts in the probability of extreme events. While STT and TME forecasts match verifications quite well in terms of spatial structure and anomaly sign, the number of anomalous transport days is underestimated compared to observations. The underestimation of the number of anomalous transport days exhibits a strong seasonal cycle, which becomes progressively worse as spring progresses into summer.

## 1 Introduction

Mass transport is important to many aspects of Pacific-North American climate, including: stratosphere-to-troposphere transport (STT) of ozone to the planetary boundary layer, which has negative impacts on human health (Fiore et al. 2003; EPA US 2006; Langford et al. 2009; Lefohn et al. 2011); STT to the free troposphere, which is needed to estimate the North American background distribution of ozone (Fiore et al. 2014, Cooper et al. 2015, Young et al. 2018); and water vapor
transport, which contributes to precipitation variability (Ralph and Dettinger 2011; Mahoney et al. 2016; Guan et al. 2015; Gershunov et al. 2017). Because of these impacts, identifying time periods when transport forecasts might be skillful on subseasonal timescales (forecasts 3-6 weeks into the future) is recognized as having high societal value (e.g., Lin et al. 2015; Baggett et al. 2017 and references therein).





Skillful subseasonal transport forecasts hinge, in large part, on the skillful prediction of atmospheric teleconnections (Baggett et al. 2017; DeFlorio et al. 2019). Initial studies of subseasonal teleconnection variability suggested that enhanced predictability might occur during spring when strong El Niño-Southern Oscillation (ENSO) conditions are present (Barnston 1994; Branković et al. 1994; Branković and Palmer 1997). However, more recent studies suggest that, overall, teleconnections (Wang and Robertson 2019) and transport (DeFlorio et al. 2019) on subseasonal timescales tend to be most predictable during winter,

though precipitation forecasts over the Pacific Northwest exhibit elevated skill during spring (Wang and Robertson 2019). These results are not necessarily in conflict though, because the seasonal cycle of Pacific teleconnection patterns between winter and summer are very sensitive to the location and scale of tropical forcing (Newman and Sardeshmukh 1998; Barsugli and Sardeshmukh 2002). This suggests that during spring, year-to-year variability in the forecast signal-to-noise ratio (Kumar et al. 2000, 2007; Sardeshmukh et al. 2000; Straus et al. 2003; Wu et al. 2006), and hence Pacific basin transport and circulation

forecast skill (Albers and Newman 2019), has the potential to be quite large. Nevertheless, even if teleconnections and transport are more predictable during winter on average, predicting atmospheric circulation during spring is important in the context of stratosphere-to-troposphere transport of ozone (Lefohn et al. 2011; Lin et al. 2015) and water vapor transport (Lee et al. 2014), which together, motivate this study.

Stratosphere-to-troposphere transport and water vapor transport occur via distinct physical pathways. In midlatitudes, STT occurs mainly via two mechanisms: stratospheric intrusions, which includes tropopause folds and potential vorticity streamers (Reed and Danielson 1958; Hoerling et al. 1993; Langford and Reid 1998; Shapiro 1980); and transverse circulations in jet exit regions (Langford et al. 1998; Langford 1999). Water vapor transport events, for example those embodied by so-called 'atmospheric rivers' (Zhu and Newell 1998), also arise via two distinct processes: long-range (Lagrangian) tropical-to-

extratropical moisture exports/transport (Stohl and James 2005; Bao et al. 2006; Knippertz and Wernli 2010; Knippertz et al. 2013); and extratropical transport related to local frontal dynamics (Newman et al. 2012; Madonna et al. 2014; Pfahl et al. 2014). In this study, we focus on spring season STT that extends downwards to the mid-troposphere and planetary boundary layer (PBL), and long-range tropical-to-extratropical water vapor transports, hereafter referred to tropical moisture exports (TME).


    STT and TME have very different seasonal cycles in terms of timing and geography, which is readily observed in monthly mean climatologies (Fig. 1; see Section 2 for a detailed description of STT and TME, which are both taken from the database of Sprenger et al. 2017). Over western North America, STT of mass (and ozone) that reaches the PBL peaks in spring (Fig. 1, left column; see also, Škerlak et al. 2014; Albers et al. 2018 and references therein). Despite the strong storm track located

over the North Pacific, deep STT is limited over the ocean due to a shallow marine boundary layer. In contrast, STT of mass extending downwards into the middle troposphere (500 hPa), peaks during January and February and then slowly decreases thereafter (Fig. 1, middle column). TME also undergoes a seemingly smooth transition during winter and spring, with an initial peak extending from Hawaii to the western US during February, followed by a slow recession of transport westward, whereby





a secondary peak occurs near Japan during May (Fig. 1, third column; see also Knippertz and Wernli 2010; Mundhenk et al.

2016; Gershunov et al. 2017). The different regional and temporal characteristics of the STT and TME seasonal cycles shown in Fig. 1 are in part a reflection of the different physical processes that govern them, as outlined above. However, at least a portion of STT and TME seasonality and variability are linked by one important commonality: they are both directly modulated by large-scale Rossby waves (e.g., Ryoo et al. 2013; Albers et al. 2018), which themselves owe their propagation and breaking patterns to the strength and location of the subtropical and polar front jets (Hoskins and Ambrizzi 1993; Scott and Cammas

2002; Abatzoglou and Magnusdottir 2006; Hitchman and Huesmann 2007; Mundhenk et al. 2016; Olsen et al. 2019). For example, high TME is often observed on the western edge of blocking anticyclones in the North Pacific, where air is rising (Mundhenk et al. 2016), while STT occurs east of the block, where sinking air and stratospheric PV intrusions frequently develop (Sprenger et al. 2007). This means that the variability, and as we will show, the predictability, of both types of transport are dependent on the seasonal cycle of the Pacific jet.


Sometime between early March and late April, the Pacific jet undergoes a transition – which typically occurs very abruptly – from being strong and largely zonally contiguous between Asia and North America to being weak, with a discontinuity in the jet that spans most of the Pacific basin (Nakamura 1992; Newman and Sardeshmukh 1998; Hoskins and Hodges 2019; Breeden et al. 2020). The characteristics of this transition, and its relationship to forms of low-frequency variability that might be

predictable on subseasonal timescales (e.g., ENSO) have been explored in the context of STT of mass and ozone. For example, Breeden et al. 2020 demonstrated that early season jet transitions (mid-to-late March), which are more common during La Niña conditions, are characterized by enhanced mass transport to the PBL (see also, Lin et al. 2015 and references therein). Conversely, late transitions (mid-to-late April) have weaker transport to the PBL although the association to El Niño is somewhat weaker. However, these analyses are retrospective, and it remains unclear whether forcings such as ENSO – and

the resulting teleconnections – are actually forecast well enough to be useful when making subseasonal transport predictions.

While the predictability of mass transport on daily timescales is typically limited to less than two weeks (Lavers et al. 2016; DeFlorio et al. 2018), weekly averages of dynamical variables can occasionally have skill out to 3-6 weeks (e.g., Wang and Robertson 2019; Buizza and Leutbecher 2015; Albers and Newman 2019). This evokes the possibility that forecasts of

atmospheric transport, which may be harder for models to explicitly predict on subseasonal timescales, might be successfully inferred from forecasts of more predictable or better constrained dynamical variables. Indeed, similar ideas have been successfully applied to assess the predictability of atmospheric blocking on seasonal timescales (Pavan et al. 2000) and precipitation on daily timescales (Lavers et al. 2014; Lavers et al. 2016). Here we assess the potential predictability of transport during spring based on the predictability of zonal wind variance associated with the Pacific jet. We do so by considering a

very simple conditional probability: if 200 hPa zonal winds have a high (positive or negative) loading on a particular 200 hPa Pacific basin zonal wind pattern, then what will the corresponding shift in the probability of STT or TME be during those time periods? We first answer this question in the context of a retrospective analysis, which allows us to understand the regionality




of STT and TME for different jet patterns. Then, using the retrospective results as a guide, we utilize zonal wind hindcasts from the European Centre for Medium-Range Weather to test whether STT and TME over specific geographic regions may

be predictable for subseasonal forecast leads (weeks 3-6). For both analyses, STT and TME are taken from the ETH-Zürich Feature-based climatology database (Sprenger et al. 2017), which allows us to apply a single, self-consistent measure of transport for both the retrospective (1979-2016) and hindcast (1997-2016) analysis periods.

## 2 Pacific jet and transport data

### 2.1 Jet variability

Jet variability over the Pacific-North American region is represented via empirical-orthogonal functions (EOFs), which are based on ERA-Interim (Dee et al. 2011) monthly mean (March-May, MAM) anomalies of 200 hPa zonal wind (cosine latitude weighted 10°-70° N and 125°-270° E) for the 1979-2016 period. Anomalies were created by removing the first four annual harmonics of the 1979-2016 daily climatology. Using monthly averages instead of daily or weekly values is motivated in part by the suggestion of Newman et al. (2012) that a large fraction of ocean-to-continent transport arises from low-frequency

variability rather than individual synoptic events. Using monthly values also significantly boosts the variance explained by the leading three EOFs to nearly 60% versus <20% for daily values (e.g., Feldstein 2000). We use a bootstrap method to test for EOF degeneracy (North et al. 1982) and find that the first three EOFs (Fig. A1), which represent 25%, 21%, and 11% of the total MAM monthly mean wind variance, are reasonably well-separated and have robust spatial patterns (see Appendix for details). Hereafter we refer to the first three EOFs (and their corresponding PC time series) as EOF1 (PC1), EOF2 (PC2), and

EOF3 (PC3).

While EOFs 1-3 are significantly correlated with several commonly used climate indices (Table 1), we make no inference that the EOF patterns represent dynamical or physical "modes" of the climate system (Monahan et al. 2009). Indeed, the significant correlations between each of our PC time series and multiple teleconnection indices indicates that the variance of our EOFs

almost certainly results from a convolution of external forcing and internal variability across multiple timescales (e.g., Straus and Shukla 2002). Evidence for this assertion can be found by noting that while EOF1 is essentially uncorrelated with the NOAA Oceanic Niño Index (ONI) (correlation of 0.16 and not significant), EOF1 is one-month lag correlated with EOF2 (correlation 0.66, significance level >95%), which is itself highly correlated with the ONI index (correlation 0.78, significance level >95%). Thus, with one exception (considered in the Discussion) we simply use the EOFs as a data compression tool that

helps to isolate the largest scale flow patterns that we anticipate will have the best chance for prediction.

To evaluate the potential predictability of Pacific jet variability, we use hindcasts (1997-2016) of 200 hPa zonal wind from the European Centre for Medium-Range Weather Forecasting Integrated Forecasting System (ECMWF IFS CY43R1/R3, model operational in 2017), which were obtained from the Subseasonal-to-Seasonal Prediction Project database (Vitart et al. 2017).



Hindcasts are 'coarse-grained' in time via a 7-day running-mean and in space via regridding to a fixed 2.5-degree
longitude/latitude grid. Anomalies are computed by removing the lead dependent climatology, which also serves as a mean
bias correction (e.g., Buizza and Leutbecher 2015; Monhart et al. 2017). Hindcasts are computed as three-week averages for
weeks 3-5 (i.e., days 15-35). The 3-week averages are then projected onto the EOF patterns described above. We also computed
results for other averaging periods including weeks 3-4 and 3-6, as well as individual week 3, 4, and 5 forecasts, but settled on
weeks 3-5, because we found this window provided the best balance between predictability (averaging longer periods typically
extends the 'forecast skill horizon', e.g., Younas and Tang 2013; Buizza and Leutbecher 2015) versus extending the forecast
window out all the way to week 6 where there is very little amplitude remaining in the forecasts. The IFS hindcast PC time
series are verified against ERA-Interim-based PC time series prepared in an identical manner.

To help verify that the zonal wind EOF patterns are highlighting Pacific jet variability, we compare the EOFs to a upper
tropospheric jet stream climatology (Koch et al. 2006; Sprenger et al. 2017), which is itself based on ERA-Interim. The jet
climatology (1979-2014) is based upon vertical averaging of zonal and meridional winds between 100-500 hPa at every
horizontal grid point, where a 'jet event' at each grid point is detected when the vertically averaged wind exceeds 30 ms$^{-1}$. This
procedure yields a frequency of upper tropospheric jet 'events' at each grid point.

## 2.2 Transport composites

To examine stratosphere-to-troposphere mass transport and tropical-to-extratropical water vapor transport, we use three ETH-
Zürich Feature-based ERA-Interim Climatologies (Sprenger et al. 2017): a climatology of stratosphere-to-troposphere
transport to 500 hPa (STT$_{500}$), which provides an estimation of transport into the free troposphere;  a climatology of
stratosphere-to-troposphere transport to the planetary boundary layer (STT$_{PBL}$) (Sprenger et al. 2003, Škerlak et al. 2014); and
a climatology of tropical-to-extratropical moisture/water vapor transport (TME), (Knippertz and Wernli 2010). The STT
climatologies (1979-2016) are based on Lagrangian parcel trajectories calculated using the LAGRANTO Lagrangian transport
model (Wernli and Davies 1997; Sprenger and Wernli 2015), where stratosphere-to-troposphere mass trajectories are
considered as exchange 'events' if they have 48-hour stratospheric, followed by 48-hour tropospheric, residence times. We
use both monthly mean and daily mean climatologies of STT$_{500}$ and STT$_{PBL}$, all of which have units of number of mass
exchange events per 6-hourly time step. TME climatologies (1979-2016) are calculated via LAGRANTO water mass
trajectories that originate in the tropics and reach at least 35° N with a water mass flux greater than 100 g kg$^{-1}$ m s$^{-1}$; we use
monthly mean and daily mean TME climatologies where units are given as the number of TME events per 6-hourly time step.

For our retrospective transport analysis, we composite STT and TME for months when the zonal-wind PC time series were
larger than 1 STD. For the hindcasts, we use a slightly weaker 0.8 STD threshold in order to boost the number of samples
given the relatively short length of the subseasonal-to-seasonal hindcast database (1997-2016). We chose to keep the STD
threshold as high as possible though, because higher amplitude anomalies likely correspond to periods of higher forecast skill



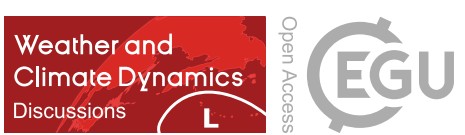

(Compo and Sardeshmukh 2004; Van den Dool and Toth 1991; Johansson 2007). Importantly, the choice of threshold does not qualitatively change the results. Hindcast transport composites are based on time periods when weekly average *forecasts* of zonal-wind PC time series were predicted to exceed 0.8 STD. For hindcast verification composites, the composites are based on periods when the *verification* weekly average zonal-winds PC time series exceeded 0.8 STD. This procedure typically means that the verification composites include more samples because, as we will show, the weeks 3-5 IFS forecasts

systematically underestimate the amplitude of the zonal wind PC time series and thus do not exceed the STD threshold as often as is observed.

While the original units of all of the ETH-Zürich climatologies are frequencies (jet frequency, STT, and TME), all of our figures, except for the climatologies (Fig. 1), are presented in units of standard deviations. That is, for every variable, we calculate anomalies from climatology and then divide by the anomaly standard deviation (z-scoring). Thus, a unit of '1 STD'

equates to a one standard deviation anomaly, where the standard deviation is calculated individually for each specific time period considered (e.g., the STD normalization for a March monthly mean is different from the STD normalization used for a three-week forecast period in March).

When comparing forecast and verification transport probability density functions (PDFs), we evaluate significance via a combination of bootstrap confidence intervals (10,000 ensembles with replacement) and two-sample Kolmogorov-Smirnov

distribution tests (KS-test; Marsaglia et al. 2003; Hollander et al. 2013), where the later tests whether the shape and location of two empirical distributions are significantly different. The PDFs themselves are created by taking box-area means of STT or TME for a specified geographic region at every forecast time step and using each as a 'sample'. The PDFs are then calculated via kernel density estimation based on the collection of all samples for either the forecasts or verifications.

## 3 Results

### 3.1 Retrospective analysis


The first three EOF patterns of the 200 hPa zonal wind all exhibit anomalies that correspond to some amount of extension/retraction and/or latitudinal shifting of the Pacific jet compared to climatology (Fig. 2). This interpretation is confirmed by compositing ETH feature-based jet frequencies for time periods with high EOF loading (PC amplitude >1 STD), which yields jet frequency distributions that correspond extremely well with each of the first three EOF wind patterns

(Supplementary Fig. S1). This suggests that the amount of wind variance explained by each of the individual EOFs is sufficiently large that when the PC loading is high there are notable, vertically deep (at least 100-500 hPa), corresponding shifts in the location of the Pacific jet stream. While the EOF patterns likely combine jet variability due to both the subtropical and polar front jets (Koch et al. 2006), a strong jet stream of either type will act as a waveguide for Rossby waves (e.g.,



Schwierz et al. 2004; Rivière 2010 and references therein) with an increased frequency of STT and TME (e.g., Shapiro and
Keyser 1990, Koch et al. 2006).

To evaluate the jet-transport connection, we consider STT and TME for time periods with high zonal wind EOF loading
(absolute value of PCs>1 STD). Because the patterns of the STT and TME anomaly composites are so similar for both EOF
phases, we show only the positive EOF pattern; see Supplement Figs. S2-S4 for the negative phase. $STT_{500}$ maxima match the
EOF wind patterns quite well (Fig. 3, top row), with positive (negative) $STT_{500}$ anomalies tending to occur along the northern
flanks of the regions of stronger (weaker) winds (Koch et al. 2006), and hence increased (decreased) jet frequency. The
correspondence of higher $STT_{500}$ with higher windspeeds, suggests that transverse circulations around the jet play a key role
in transport, and confirms that the EOF-based $STT_{500}$ anomalies are related to variations in the North Pacific storm track
(Škerlak et al. 2014). $STT_{PBL}$ on the other hand (Fig. 3 middle row), have maxima slightly downstream of the 500 hPa maxima,
which reflects the fact that deep STT tends to occur as maturing Rossby waves amplify and tropopause folds and potential
vorticity streamers extend downwards towards the surface (Wernli and Bourqui 2002; Sprenger et al. 2003, Appenzeller et al.
1996, Wernli and Sprenger 2007, Škerlak et al. 2015). Anomalous TME also corresponds well with the EOF patterns (Fig. 3,
bottom row), except that the anomalies are on the southern edge of the positive EOF wind patterns, which is due to the tendency
for strong TME to occur along the warm sector of a breaking Rossby wave (Bao et al. 2006; Knippertz et al. 2013).


While all of the transport composites are physically consistent with the EOF patterns, and hence jet variability, the $STT_{500}$ and
TME composites have a much more robust signal compared $STT_{PBL}$. That the $STT_{PBL}$ is weaker is not entirely surprising
because while a high percentage of upper level breaking waves extend downwards to the mid- to upper troposphere,
subsequently causing associated $STT_{500}$ and TME, only a small subset of these waves will achieve the needed amplitude and
depth to extend all the way to the PBL. Moreover, transport to the $STT_{PBL}$ is also dependent on the depth of the PBL, which
tends to be relatively shallow until late spring to early summer when convective heating begins to increase (Seidel et al. 2012;
Škerlak et al. 2014; Breeden et al. 2020). Nevertheless, all of the composites provide a basis for the expectation that Pacific
jet variability can be used as a predictor for transport over landmasses of interest, including the western United States, southern
Alaska, and Japan.

**3.2 Potential predictability of jet shifts and transport**

While subseasonal forecasts of teleconnection indices are known to exhibit reasonable correlation-based skill (Wang and
Robertson 2019), the amplitude of the anomalies is often quite weak compared to observations (Yamagami and Matsueda
2020). Thus, the relevant question here is, do forecast models predict jet variability well enough – in terms of both correlation
and amplitude  –  to provide guidance for subseasonal transport forecasting?






For weekly forecasts, the correlation between the forecasted and verified zonal wind PCs is 'skillful' (correlations >0.5-0.6, Hollingsworth et al. 1980; Arpe et al. 1985; Murphy and Epstein 1989) within the deterministic timeframe (weeks 1-2) for all three EOFs (Table 2). Beyond week 2, however, the PC1 and PC3 correlations drop off rapidly, with the skill of predicting PC3 almost completely limited to synoptic timescales. On the other hand, PC2 retains useful skill all the way out to forecast

week 6, which may be due to its stronger relationship to ENSO (Table 1). These correlations suggest that only the first two PCs retain enough skill to be useful on subseasonal leads. The same result is true for the weeks 3-5 forecast window (Fig. 4), where forecast-verification correlations for both PC1 and PC2 are near or above 0.5, while PC3 exhibits very low correlation-based skill. In terms of the PC amplitudes of the weeks 3-5 forecasts, both PC1 and PC2 regularly exceed our 0.8 STD threshold, while the PC3 amplitude rarely exceeds it. Thus, while EOF3 is related to large transport anomalies over land regions

of interest (e.g., $STT_{PBL}$ and TME over North America), it is unfortunately not predictable on subseasonal timescales (similar results are also found for EOFs 4 and higher). We therefore focus on predicting transport via PC1 and PC2.

The number of observed instances (verifications) when the PC1 and PC2 amplitudes exceeds 0.8 STDs exhibits a seasonal cycle (Fig. 5), though the degree of overlap of the confidence intervals suggests that the seasonal cycle is more pronounced

for EOF2 than for EOF1. The situation is a bit more complicated if the individual phases of each EOF are considered (see Supplement, Fig. S5), though the small sample sizes make conclusive inferences difficult. Nevertheless, the slow decay of observed PC1 and PC2 exceedances (i.e., large amplitude jet events) between March and May is qualitatively consistent with previous studies documenting the seasonality of jet activity and Pacific baroclinic wave amplitudes (Nakamura 1992; Koch et al. 2006). Unfortunately, the number of PC1 and PC2 exceedances predicted by the IFS at 3-5 week lead times has a much

stronger seasonal cycle compared to observations, with early spring having many more exceedances than for late spring for both phases of PC1 and PC2 (Fig. 5 and Supplement S5). This implies that the transport anomalies outlined next are more predictable, and hence the composites more heavily weighted, for the periods before the jet undergoes its spring transition (Newman and Sardeshmukh 1998; Breeden et al. 2020).

Based on the regions with the largest transport anomalies (Fig. 3) for the more predictable PC1 and PC2 time series (Fig. 4), we chose four subregions within the full Pacific domain to examine the potential predictability of STT and TME: EOF1-based $STT_{500}$ for the North Pacific, which includes southern Alaska and the Russian Far East; EOF2-based $STT_{PBL}$ for the western to intermountain-western US; EOF1-based TME for the western US; and EOF2-based TME for the West Pacific (Japan and far eastern Asia). These subregions are highlighted by the boxes in Figs. 6a, 7a, 8a, and 9a, respectively. To provide context

for the four subregion forecasts, we first show forecast and verification transport anomalies for the entire Pacific domain. For each of the four full domain figures (Figs. 6-9), the top two panels show verification transport composites, which are based on times when the *verification* zonal wind PC time series amplitude is greater than +/-0.8 STD (black lines in Fig. 4), while the bottom two panels show corresponding transport composites, except for time periods when the *forecasted* zonal wind PC time series amplitude is greater than +/-0.8 STD (orange lines in Fig. 4). For comparison, the months that are included in the



retrospective composites (Fig. 3) are highlighted by the light red and blue shading in Fig. 4 (note that the time periods when the week 3-5 time series exceed the +/-0.8 STD threshold do not always match the red and blue shading regions, because the shaded regions highlight periods when the monthly mean time series exceeded the monthly 1 STD threshold). The pattern correlations between the forecast and verification transport composites for the full domains in Figs. 6-9 (not just for the boxed in areas) are included in the forecast titles for both EOF phases. Transport predictability for the four boxed subregions is

subsequently evaluated via PDFs of transport for the forecasts and verifications (Fig. 10).

$STT_{500}$ based on the EOF1 forecast is qualitatively consistent with the verification-based composite for both positive and negative EOF phases (Fig. 6), though the $STT_{500}$ pattern is better reproduced for the negative phase (pattern correlation of 0.49 vs. 0.75 for the positive vs. negative phases, respectively). In addition, the verification composites show an asymmetry between

opposite EOF1 phases in the amount of $STT_{500}$, which is also accurately forecasted, with the negative EOF phase exhibiting peak values in the 0.75-1.25 STD range vs. 0.25-0.5 STDs for the positive EOF phase. This asymmetry is likewise reflected in the forecast and verification PDFs of $STT_{500}$ for the North Pacific subregion (Fig. 10a), where the median for the positive EOF1 phase is weakly negative, while the negative EOF1 phase has a greater than +0.5 STD median anomaly. The only noteworthy difference between the North Pacific forecast and verification PDFs is that the forecast-based PDF is shifted

towards more positive values than the verification-based PDF. Regardless, the confidence intervals for the medians of the positive vs. negative phases of the forecast-based PDFs are very well-separated and the underlying distributions are different according to a KS-test, which suggests that the predicted shifts in transport are significant. We also evaluated forecasts of $STT_{500}$ for various subregions over populated land masses (e.g., the western US), but the resulting verification and forecast PDFs were not significantly different, which reflects the fact that $STT_{500}$ peaks over the North Pacific portion of the storm

track (Fig. 3a).

For EOF2-based $STT_{PBL}$ over the western US, the verification composite is consistent with the retrospective composites (cf., Figs. 7a,b and Fig. 3e), however, the pattern is much weaker. One potential reason that the forecast and verification patterns are weaker than in the retrospective analysis is the smaller time averaging window used when constructing the forecast and

verification composites (3 week averaging windows for the forecast/verification composites vs. 4 week averaging windows for the retrospective composites). However, if the forecast averaging window is expanded to 4 weeks, fewer well-forecasted periods are included in the composites, which also leads to a weaker $STT_{PBL}$ pattern. Nevertheless, the forecast- and verification-based $STT_{PBL}$ composites (Fig. 7c,d) and PDFs for the western US subregion (Fig. 10b) do agree quite well, though the $STT_{PBL}$ distribution is notably shifted away from zero only for the negative EOF phase. The confidence intervals for the

medians overlap; thus, on average, the $STT_{PBL}$ forecasts appear to be borderline in their usefulness. Still, the forecasted $STT_{PBL}$ do represent different distributions according to a KS-test, so the change in the shape of the tails of the distributions may be of some practical use for prediction of extreme $STT_{PBL}$ events.



The TME forecasts match the verifications very well, for both EOFs, and for both phases of each EOF, with basin-wide pattern correlations ranging from 0.57 to 0.88 (Figs. 8-9). In addition, the magnitude of the anomaly values for both EOFs are notable, with both TME phases exhibiting anomalies in the 0.5-1.25 STD range over relatively large portions of the Pacific domain. Interestingly, positive TME centered over Alaska is predicted very well for the positive phase of EOF1 (Fig. 8a,c) and the negative phase of EOF2 (Figs. 9b,d), yet it is unclear if this pattern represents a reliably predictable form of TME because neither of the corresponding TME composites for the longer time record retrospective analysis show anomalies over Alaska (cf., Figs. 8 and 9 to Figs. 3g and 3h, respectively). In contrast, the forecasted patterns of TME between Japan and the west coast of the US (south of 55º N) are quite consistent with the jet (Figs. 1 and S1) and TME (Fig. 3, bottom row) patterns from the retrospective analysis, which suggests that TME over broad regions of the Pacific basin may be reasonably predictable during spring. Indeed, the western US and West Pacific subregion TME PDF shifts are robust and match the verification PDFs very well (Fig. 10c and d, respectively). This is particularly true for the West Pacific where the median shift in TME transport is nearly +/- 1 STD for each EOF phase, and the PDF forecast and verification PDFs are nearly identical.

## Discussion and conclusions

Many 'modes' of climate variability are known to be associated with anomalous atmospheric transport. For example, stratosphere-to-troposphere mass and ozone transport to the PBL over North America is known to be influenced by ENSO (Breeden et al. 2020; Lin et al. 2015 and references therein), while the frequency of atmospheric rivers is thought to be modulated by a variety of climate phenomena, including ENSO, the Madden-Julian oscillation, and the quasi-biennial oscillation (Guan et al. 2012; Lee et al. 2014; Kim and Alexander 2015; Guan et al. 2015; Mundhenk et al. 2016; Guirguis et al. 2019). However, retrospectively isolating such associations, which is equivalent to conducting a 'perfect model' forecast, does not assure that current operational forecast models can successfully predict those relationships, particularly on subseasonal timescales (e.g., Lavers et al. 2016; Baggett et al. 2017). Nevertheless, some teleconnection and transport patterns appear to be potentially predictable on subseasonal timescales (e.g., Mundhenk et al. 2018; Wang and Robertson 2019; Pan et al. 2019; DeFlorio et al. 2019; Yamagami and Matsueda 2020), though these forecasts are typically found to occur during boreal winter.

Our analyses have shown that stratosphere-to-troposphere transport (STT) to at least 500 hPa and long-range tropical-to-extratropical moisture transport (TME) over the Pacific-North American region can potentially be skillfully predicted on subseasonal timescales (3-5 weeks ahead of time) during boreal spring. The transport forecasts themselves were inferred from ECWMF IFS-based forecasts of Pacific jet variability. IFS Pacific jet forecasts for four Pacific-North American subregions are associated with significant shifts in the probability of anomalous transport, including: STT into the free troposphere over the North Pacific (Fig. 10a); STT into the planetary boundary layer over the intermountain-western US (Fig. 10b); TME over the west coast of the US (Fig. 10c); and TME to Japan and far eastern Asia (Fig. 10d). While the forecasted shifts in transport





probability match verifications quite well, one deficiency is apparent: the IFS is able to predict the sign of the zonal wind PC time series with reasonable success (Table 2 and Fig. 4), yet it consistently struggles to maintain enough zonal-wind PC amplitude relative to the substantial weather-related noise (compare amplitude of forecast and verification time series in Fig. 4). This results in an underestimation of the number of anomalous transport days compared to observations (Fig. 5), which

degrades the estimation of the transport probabilities (Fig. 10).

The underestimation of the number of anomalous transport days exhibits a strong seasonal dependence, which becomes quite acute during April and May (Fig. 5). This implies that either overall teleconnection predictability decreases as spring proceeds, or alternatively, the IFS is simply unable to skillfully predict large amplitude jet anomalies with consistency beyond early-

spring. While it is beyond the scope of the current study to explore which one of these possibilities is responsible for the lack of consistent late spring skill, this is clearly an important question, because the first possibility would be a fundamental feature of the climate system, while the latter would be a model-based constraint that might theoretically be improved. Of course, these two possibilities are not mutually exclusive, because the increasing sensitivity of Pacific-North American teleconnections to tropical forcing at smaller spatial scales during the spring jet transition (Newman and Sardeshmukh 1998) may be inherently

less predictable, yet also more difficult to accurately model. That said, despite the IFS underestimation of the number of days with anomalously strong jet patterns (Fig. 5), the IFS is still able to identify roughly 15% (PC1) and 30% (PC2) of all spring days (March-May) that are anomalous, which suggests that using upper-level winds to forecast transport may currently be possible.

For the three types of transport that we have evaluated here, STT into the free troposphere and TME are the most robustly predicted, at least in terms of shifts of the average and extremes of their transport distributions (Fig. 10). STT to the PBL over the western US, on the other hand, mainly exhibits a change in the shape of the tails of the transport distributions, but a rather weak shift in the median (i.e., the shift of the medians of the two EOF2 phases have confidence intervals that are strongly overlapping, Fig. 10b). This has implications for the suggestion that ENSO may be used to predict air quality related to STT

of ozone during spring (e.g., Lin et al. 2015 and Albers et al. 2018 and references therein). Similar to previous retrospective analyses (e.g., Lin et al. 2015; Breeden et al. 2020), we find that mass transport to the PBL is associated with ENSO (Fig. 11), where here, we have composited $STT_{PBL}$ based on periods when the NOAA ONI is greater than 0.8 STDs from the historical mean, which yields an equivalent number of samples to our EOF2-based results. The ONI-based (retrospective) transport composites look very similar to our earlier EOF2-based retrospective results (cf., Figs. 11a and b to Figs. 3e and S3c,

respectively. For proper comparison, note that PC2 and ONI are negatively correlated). Moreover, the transport PDFs for the intermountain-western US subregion based on PC2 versus ONI, for both ENSO phases, are drawn from the same distributions according to a two-sample Kolmogorov-Smirnov test (Fig. 11c). This close correspondence is due to the high correlation between the ONI and PC2 time series (Fig. 11d). Yet, because we have found $STT_{PBL}$ predictions related to EOF2 to be significant only in terms of shifts in the tails of the distributions (cf. Fig. 10b and 11c), our results suggest that at best, ENSO

may be harnessed to provide STT$_{PBL}$ forecast guidance on subseasonal timescales for extreme events only. Complicating matters further in the context of ozone transport to the PBL (as opposed to simply mass transport as investigated here), is that predictions based on ENSO will likely be even more difficult because STT of ozone is also modulated by the seasonal variability of the available reservoir of ozone in the extratropical lower stratosphere (Neu et al. 2014; Albers et al. 2018). That said, because it is doubtful that Nino-3.4-based indices like ONI capture the full dynamical scope of ENSO variability (Penland

and Matrosova 2006; Capotondi et al. 2015), the complete impact of ENSO on STT$_{PBL}$ predictability certainly deserves further study.

**Data Availability**

The ERA-Interim reanalysis data used in this study is available through the National Center for Atmospheric Research Consortium for Atmospheric Research Data Archive: https://rda.ucar.edu/. The STT, TME, and jet ERA-Interim feature-based

climatology data is available from http://eraiclim.ethz.ch/. ECMWF IFS hindcast data is available via the S2S Prediction Project (http://s2sprediction.net/).

**Author Contributions**

John R. Albers wrote retrospective and hindcast analysis code, created the figures, and wrote the manuscript. Amy H. Butler, Melissa L. Breeden, Andrew O. Langford, and George N. Kiladis provided comments and edited the manuscript.

**Competing Interests**

The authors declare that they have no conflict of interest.

**Acknowledgements**

The authors would like to thank Michael Sprenger for graciously making the 6-hourly ETH Feature-based data available, which made this study possible. JRA and AHB were funded in part by NSF grant #1756958. MLB was funded by NOAA

Climate and Global Change Postdoctoral Fellowship Program, administered by UCAR's Cooperative Programs for the Advancement of Earth System Science (CPAESS) under award # NA18NWS4620043B. The authors would like to thank Yan Wang for preparing the IFS S2S data and Benjamin Moore, who helped furnish the 6-hourly TME data.

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





|  | WP | PNA | ONI |
|---|---|---|---|
| **PC1** | -0.66 (0.00) | 0.44 (0.00) | 0.16 (0.1) |
| **PC2** | -0.38 (0.00) | -0.09 (0.35) | -0.78 (0.00) |
| **PC3** | -0.31 (0.00) | -0.56 (0.00) | -0.05 (0.64) |

**Table 1:** Correlations between MAM monthly average PC time series and various climate indices, with p-values in parentheses. The West
Pacific pattern (WP) and Pacific-North American pattern (PNA) and NOAA Oceanic Niño Index (ONI) are taken from NOAA Center for
Weather and Climate Prediction (NOAA CPC).





| | *week 1* | *week 2* | *week 3* | *week 4* | *week 5* | *week 6* |
|---|---|---|---|---|---|---|
| **PC 1** | 0.96 | 0.78 | 0.56 | 0.42 | 0.3 | 0.31 |
| **PC 2** | 0.97 | 0.86 | 0.73 | 0.7 | 0.67 | 0.66 |
| **PC 3** | 0.94 | 0.67 | 0.38 | 0.21 | 0.11 | 0.09 |

**Table 2:** Correlations between MAM weekly average PC time series of IFS hindcasts and ERA-Interim verifications.

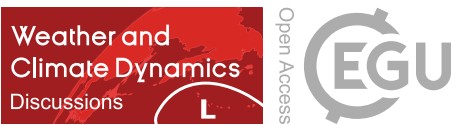

**Figure 1:** Monthly average climatologies (1979-2014) of STT to the PBL (left column), STT to 500 hPa (middle column), and TME (right column). Units for all panels are event frequencies (events/6-hourly time step), where each of the relevant events types are defined in Sect. 2.2.




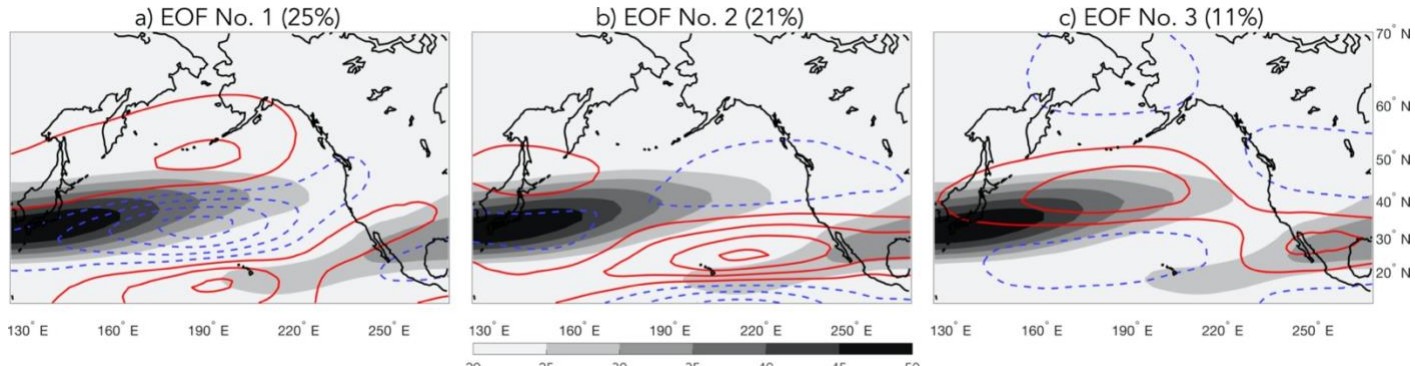

**Figure 2:** Spring (MAM, 1979-2014) zonal wind climatology (filled contours) with colored contours showing the first three EOF patterns. The variance explained by each EOF is shown in the title for each panel.





**Figure 3:** Monthly mean (MAM, 1979-2014) frequencies (filled contours) of STT to 500 hPa (top row), STT to the PBL (middle row), and TME (bottom row), for time periods when PCs 1-3 are greater than +/-1 STDs from climatology (units of STDs). Colored contours show the EOF patterns associated with each composite.

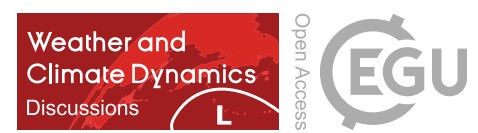

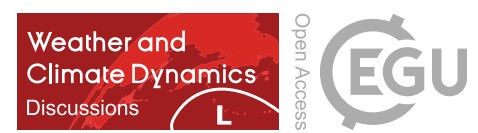

**Figure 4:** Time series of weeks 3-5 average zonal wind projected onto EOFs 1-3 for IFS forecasts (orange lines) and ERA-Interim verifications (black lines). The horizontal dashed lines denote +/-0.8 STDs from the mean of the verification time series. For reference, the light blue and red shading denote the months that were included in the monthly average composites used to create Fig. 3. Correlations between the forecasts and verifications (with 95[th] percentile confidence intervals) are shown in the titles of each panel.

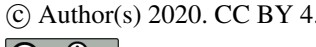



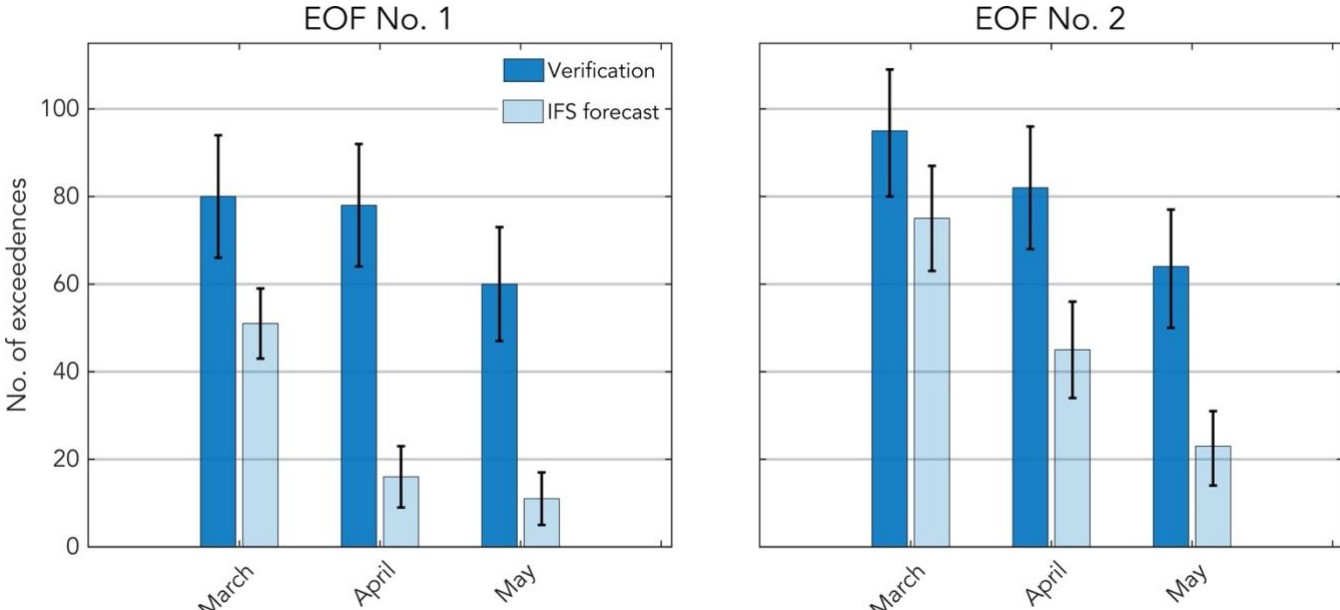

**Figure 5:** Number of times that a weeks 3-5 average verification or forecast exceeded the 0.8 STD threshold for the 1997-2016 hindcast period (i.e., the periods in Fig. 4 where the black or orange lines, respectively, was above or below the dashed horizontal STD reference lines). 95th percentile bootstrap confidence intervals are shown as whiskers.



**Figure 6:** (a), (b) EOF1-based composites of STT to 500 hPa for weeks 3-5 forecast periods when the verification time series (black line in Fig. 4) was above (positive phase) or below (negative phase) the 0.8 STD threshold. (c), (d) EOF1-based composites of STT to 500 hPa for weeks 3-5 forecast periods when the forecast time series (orange line in Fig. 4) was above (positive phase) or below (negative phase) the 0.8 STD threshold. The black box outlines the North Pacific subregion used for creating the transport PDF in Fig. 10a. Units are in STDs and pattern correlations between top and bottom panels (cf., (a) versus (c) and (b) versus (d)) are shown in the bottom row titles.



**Figure 7:** (a), (b) EOF2-based composites of STT to the PBL for weeks 3-5 forecast periods when the verification time series (black line in Fig. 4) was above (positive phase) or below (negative phase) the 0.8 STD threshold. (c), (d) EOF2-based composites of STT to the PBL for weeks 3-5 forecast periods when the forecast time series (orange line in Fig. 4) was above (positive phase) or below (negative phase) the 0.8 STD threshold. The black box outlines the western to intermountain-western US subregion used for creating the transport PDF in Fig. 10b. Units are in STDs and pattern correlations between top and bottom panels (cf., (a) versus (c) and (b) versus (d)) are shown in the bottom row titles.

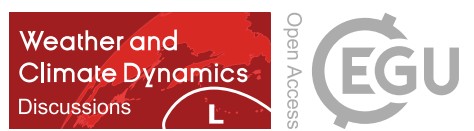

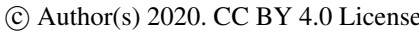

**Figure 8:** (a), (b) EOF1-based composites of TME for weeks 3-5 forecast periods when the verification time series (black line in Fig. 4) was above (positive phase) or below (negative phase) the 0.8 STD threshold. (c), (d) EOF1-based composites of TME for weeks 3-5 forecast periods when the forecast time series (orange line in Fig. 4) was above (positive phase) or below (negative phase) the 0.8 STD threshold. The black box outlines the western US subregion used for creating the transport PDF in Fig. 10c. Units are in STDs and pattern correlations between top and bottom panels (cf., (a) versus (c) and (b) versus (d)) are shown in the bottom row titles.

**Figure 9:** (a), (b) EOF2-based composites of TME for weeks 3-5 forecast periods when the verification time series (black line in Fig. 4) was above (positive phase) or below (negative phase) the 0.8 STD threshold. (c), (d) EOF2-based composites of TME for weeks 3-5 forecast periods when the forecast time series (orange line in Fig. 4) was above (positive phase) or below (negative phase) the 0.8 STD threshold. The black box outlines the West Pacific subregion used for creating the transport PDF in Fig. 10d. Units are in STDs and pattern correlations between top and bottom panels (cf., (a) versus (c) and (b) versus (d)) are shown in the bottom row titles.







**Figure 10:** Probability density functions (PDFs) of (a) EOF1-based STT to 500hPa for the North Pacific subregion, (b) EOF2-based STT to the PBL for the western to intermountain-western US subregion, (c) EOF1-based TME to the western US subregion, and (d) EOF2-based TME to the West Pacific subregion. IFS-based forecasts are shown in solid dark lines (with medians shown as blue dots and 95th percentile bootstrap confidence intervals shown as whiskers) and ERA-Interim-based verifications are shown as thicker light lines. Units are in STDs.



**Figure 11:** (a) El Niño- and (b) La Niña-based monthly mean (MAM, 1979-2014) frequencies of STT to the PBL (filled contours) and zonal
winds (contours) for time periods when the NOAA ONI was +/-0.8 STDs from climatology (units of STDs). Note: for correct comparison,
panels (a) should be compared to panels (d) from Fig. 4; compare also panels (a) and (b) here to panels (d) and (c) from Fig. S3. (c) Probability
density functions (PDFs) of EOF2-based STT to the PBL for the western to intermountain-western US subregion. (d) Time series of the
NOAA ONI (blue line) and PC2 (orange line), where ONI has multiplied by -1 for ease of comparison.




**Appendix**

To verify that EOFs 1-3 represent distinct patterns that are robust to variations in sampling period (North et al. 1982), we conducted several calculations. To begin, a 10,000 member bootstrap ensemble of 200 hPa zonal wind EOFs was created (resampling with replacement), where each bootstrap member consisted of 'N' randomly selected monthly mean 200 hPa zonal

wind anomalies for the Pacific basin domain shown in Fig. 2. The 'N' randomly selected anomalies are chosen from the pool of all MAM 1979-2016 monthly means, and N=114, which is the number of months in the original EOF calculation for MAM, 1979-2016. The resulting data was used in three calculations.

First, the pattern correlation between each bootstrap ensemble member EOF and the corresponding original EOF was

calculated. The median pattern correlation for all 10,000 bootstrap ensemble members was then calculated. For all three EOFs, the median pattern correlation was near 0.9 (individual values are shown for each of the three EOFs in the title bars of Fig. A1 a-f). Next, the median of the variance explained was calculated for each bootstrap ensemble EOF. For all three EOFs, the variance explained for the original EOFs and for the median of the bootstrap ensemble EOFs is within a couple percent (individual values are shown for each of the EOFs in the title bars of Fig. A1 a-f). And finally, the standard deviation of the

variance explained was calculated for each of the bootstrap ensembles (Fig. A1g). The spread (measured by the standard deviation) is small enough that there is no overlap between each of the first three EOFs. In combination, these calculations support the notion that the first three 200 hPa zonal wind EOFs are not degenerate according to the criteria outlined in North et al. 1982.






Figure A1: (a), (c), and (e) 200 hPa zonal wind EOF patterns for MAM, 1979-2016, which correspond to the EOF contours shown in Figs. 2-3 and S1-S4. (b), (d), and (f) 200 hPa zonal wind EOF patterns for the bootstrap ensembles corresponding to panels (a), (c), and (e), respectively. For each row in (a)-(f), the median pattern correlation between the original (left column) and bootstrap ensembles (right column) are shown in the subtitle. The subtitle of each panel in (a)-(f) also shows the variance explained (original EOFs, left column) or the median variance explained (bootstrap ensembles, right column) for each EOF. (g) Median variance explained for the bootstrap ensemble (solid marker), and the spread of variance explained for the bootstrap ensembles of each EOF, where the spread is calculated as 1 STD of the variance explained (shown as whiskers).