# Peer review of "Subseasonal prediction of springtime Pacific-North American transport using upper-level wind forecasts"

_Weather and Climate Dynamics, 2020_

## Referee Comment (RC1) · Anonymous Referee #1 · 19 Jan 2021

General Comments: This work explores the use of subseasonal time scale forecasts of jet variability to predict stratosphere-to-troposphere transport and tropical-to-extratropical moisture transport over the NH Pacific and surrounding land mass. The study does a good job of first examining the transport features with the leading EOFs of the upper-level Pacific jets. The ability to use the forecasts of the upper-level winds to predict the transport types is then considered. The study is appropriate for the journal and should be of interest. Overall, the paper is fairly well constructed and written. I believe it should be acceptable for publication after the authors address a few minor issues.

Specific Comments:

1) I wonder why only the spring months were examined. I believe the seasonality of the STT would limit the ability during summer and fall, but I would think the magnitude of STT during DJF is great enough to be useful. At the very least, a short discussion of the other seasons and the possibility of prediction outside of MAM should be presented.

2) Lines 185-187: It is not clear how the claim of "vertically deep" shifts in the jet stream is made from the 2-D comparison. I am assuming that is because the jet climatology is derived from vertically averaged winds between 100-500 hPa. However, given the strong vertical gradients of wind around the jet stream, the wind changes don't have to be vertically deep to result in shifts to the jet. In addition, the phrase could also be interpreted to imply the jet stream location shifts vertically which is not necessarily the case.

3) Line 223 and Table 2: Please add the significance level information (e.g. what is >95% significance level?). The correlations are much less meaningful without this information.

4) There are several instances where the references cited only include recent papers and could benefit by being more historical. For examples: lines 201-202, previously cited work by Reed and Shapiro should be included here too; and Line 358, Olsen, et al., JGR, 2013.

5) Fig 10 and related discussion: There are multiple regions and 3 transport types considered. Figure 10 and the discussion concentrates on only 4 of these possible combinations. Are the results not statistically significant for the other area/transports that are not discussed? Or is the discussion representative of the other areas too? Some comment on this should be made in the paper.

6) Many of the figures lack labels on the color bars and/or the axes. These need to be added.

7) The color bar placements in Figure 1 should be better aligned to indicate which figures they are applicable to.

8) What are the contour levels in Figure 2?

9) Fig. 10: It would be greatly helpful if the color coding of the lines were shown in a legend. Also, why are the medians and whiskers not shown for the verifications? Please include them since it would be useful in the comparisons.

Technical Corrections:

1) Lines 138-142: Please add: "...jet variability [in Section 3.1], we compare the EOFs..." for clarity.

2) Line 192: "loading" likely more appropriately "magnitude"

---

## Referee Comment (RC2) · Anonymous Referee #2 · 30 Jan 2021

Review for

Subseasonal prediction of springtime Pacific-North American transport using upper-level wind forecasts

by Albers et al.
* * *
**Synopsis:**

In their study, Albers et al. look at subseasonal prediction of stratosphere-to-troposphere transport (STT) and tropical moisture exports (TME). In contrast to previous attempts in this direction, they rely on the upper-level wind forecasts, i.e., on the location and intensity of the jet streams in the Pacific-American sector, which they characterize by the leading three components in an EOF analysis. First, the approach is considered in a retrospective perspective, based on ERA-Interim reanalysis data. Then, hindcast simulations are used to apply it to forecasts at different lead times. The relevance of the research topic is clearly motivated in the introduction, the text is very well written and follows a very clear storyline, and the number and quality of the figures and tables well supports the scientific findings. There is no doubt that the study will be of interest to the readership of Weather and Climate Dynamics. In summary, I can recommend the publication of the study with minor changes, which are listed below. The only aspect that serves a little more attention is the specific role that is attributed to tropopause folds and PV streamers, as proxies for Rossby wave breaking events. It would be nice if the text gets more specific (lee speculative) and possibly also incorporates some feature based analysis.

**Major concerns:**

1. Not a concern, but a thank you for a very concise and very clear introduction! It is a little long, but provides a very introduction into the topic and reads very fluently.

2. Tropopause folds, Rossby wave breaking and PV streamers are mentioned at several places in the text. Their role, however, remains somewhat unclear: Some examples are:

> -L187-190: "While the EOF patterns likely combine jet variability due to both the subtropical and polar front jets (Koch et al. 2006), a strong jet stream of either type will act as a waveguide for Rossby waves (e.g., Schwierz et al. 2004; Rivière 2010 and references therein) with an increased frequency of STT and TME (e.g., Shapiro and Keyser 1990, Koch et al. 2006)"
> -L200-202: "STTPBL on the other hand (Fig. 3 middle row), have maxima slightly downstream of the 500 hPa maxima, which reflects the fact that deep STT tends to occur as maturing Rossby waves amplify and tropopause folds and potential vorticity streamers extend downwards towards the surface (Wernli and Bourqui 2002; Sprenger et al. 2003, Appenzeller et a. 1996, Wernli and Sprenger 2007, Škerlak et al. 2015)."
> -L207-210: "That the STTPBL is weaker is not entirely surprising because while a high percentage of upper level breaking waves extend downwards to the mid- to upper troposphere, subsequently causing associated STT500 and TME, only a small subset of these waves will achieve the needed amplitude and depth to extend all the way to the PBL"

My specific concerns/questions in this context are: What does it mean that a PV streamer extends down to the surface? Are TMEs are really mentioned in Shapiro and Keyer (1990) and Kock et al. 2006) to be linked to Rossby waveguides? Furthermore, the feature-based

dataset presented in Sprenger et al. (2017) also includes tropopause folds and PV streamers (as proxies for Rossby wave breaking); would it, therefore, be worthile to support the statements in this section by the corresponding composites of these features, as they are shown for TME and STT. I think that the physical argument given by the authors is essentially correct, but it would be nice to support it by the corresponding composites, if they can 'easily' be calculated.

3. Whereas the signals for STT-500hPa and TME are rather clear, it is much more difficult to see the signal of STT-PBL. For instance, in Figure 7 it is really difficult to see the signal in the west-American box for the hindcast simulations. The same applies, to a lesser degree, to the retrospective analysis in Figure 3. I wonder whether this would become somewhat clearer if (especially in Figure 7) the colorbar is adjusted. Overall, I have have the impression the link between the jet structures, as expressed in  PC1-3, is clearly discernible in for STT-500hPA and TME, but rather weak for STT-PBL. Actually, Figure 10 shows that there is some skill, but it is difficult to get it from the Figure 7.In summary, I see that the authors are aware of this fact and argue by means of Rossby waves not extending to the surface (see point 2 above) and PBL effects (altitude) to explain this weak signal of STT-PBL compared to STT-500hPA and TME, but the explanation is not fully convincing. Please add, at least, some references supporting the argument..

**Minor comments:**

-L48: As a more recent study linking STT and PV streamers, the authors might want to add the following reference:

> *Sprenger, M., Wernli, H., & Bourqui, M. (2007). Stratosphere–Troposphere Exchange and Its Relation to Potential Vorticity Streamers and Cutoffs near the Extratropical Tropopause,* Journal of the Atmospheric Sciences, *64(5), 1587-1602.*

- L51: "extratropical transport related to local frontal dynamics" This is a little too narrow, also given the references given in brackets; the references actually refer to warm conveyor belts (WCB) and, therefore, it might be appropriate to bring up this term and explain also in one, two sentences how WCBs, TMEs and atmospheric rivers are related. A possible reference could be:

> *Sodemann H. et al. (2020) Structure, Process, and Mechanism. In: Ralph F., Dettinger M., Rutz J., Waliser D. (eds) Atmospheric Rivers. Springer, Cham. https://doi.org/10.1007/978-3-030-28906-5_2*

- L60: "deep STT is limited" -> "deep STT into the PBL is limited"; to make very clear that 'deep' refers to PBL-reaching trajectories

- L103: The retrospective (1979-206) and hindcast (1997-2016) analysis periods are introduced here, which is fine. But at first reading, I wondered a little why these time periods are chosen and what exactly are the underlying datasets. This becomes very clear only in section 2.1, but would have 'helped' me already at this place.

- Figure 1: I am somewhat confused how I have to link the two color bars to the different panels. This should be mentioned, at least, in the figure caption. Furthermore, I wonder whether the unit 'events/6-hourly time step' could be replaced by an equivalent, but more intuitive 'frequency per month'?

- Figure 2: The jet variability is expressed by the first three components of an EOF analysis. This is fine, and the three EOF patterns in Figure 2 clearly show distinct jet signatures (structures). Just to be sure: Would it be correct to name EOF1 a 'double-jet structure', EOF a 'single jet structure and EOF3 a 'transitional structure'? The patterns are OK, but it would be nice to attribute them to a somewhat less abstract (EOF loadings) picture.

-L136: "is very little amplitude remaining"; What does 'amplitude' mean in this context? Would 'skill' be the better word?

- L152-153: "units of number of mass exchange events per 6-hourly time step"; It might be worthwhile in one sentence more clearly what this unit means. How is a 'mass exchange event' defined?

-L218: "in terms of both correlation and amplitude"; really 'correlation', or should it be 'location'?

---

## Author Comment (AC1) · 18 Feb 2021

"***Subseasonal prediction of springtime Pacific-North American transport using upper-level wind forecasts***"
by J. Albers et al. (WCD paper # wcd-2020-60)

We thank Reviewer #1 for their comments, suggestions, and reference suggestions, which significantly improved the manuscript. We discuss the references that we have added below.

Reviewer wrote: *I wonder why only the spring months were examined. I believe the seasonality of the STT would limit the ability during summer and fall, but I would think the magnitude of STT during DJF is great enough to be useful. At the very least, a short discussion of the other seasons and the possibility of prediction outside of MAM should be presented.*

Our response: The reviewer is certainly correct that DJF skill (at least for TME and STT to 500 mb) is likely large enough to be useful. That said, we are specifically interested in MAM for several reasons. Since the reviewer specifically asked about STT, we will discuss this first, but we follow with an overall rationale that applies to both STT and TME.

The original motivation for our study was to examine STT to the PBL over North America, because this is the type of STT that can directly influence surface air quality. For this specific type of STT, spring is by far the most important time period, which our study is not the first to point out (see for e.g. the references we included: Fiore et al. 2003; EPA US 2006; Langford et al. 2009; Lefohn et al. 2011; note that our Fig. 1-first column also supports this).

In terms of STT to 500 hPa and TME (and water vapor transport more generally) on the other hand, the reviewer is quite right that transport skill during DJF is an interesting topic. Indeed, water vapor transport is large in magnitude (e.g., atmospheric rivers from the Pacific basin to the west coast of the US) and forecast skill perhaps at a peak during DJF. This is likely one reason why there are so many studies (too many to count) that discuss the predictability of water vapor transport during DJF. However, there are very few modern studies (if any?) that have examined STT and TME forecast skill during spring, despite the fact that this is a dynamically interesting period (see for e.g., the Breeden et al. paper we cite) and is an important season for moisture transport over the Pacific-North American region (e.g., Mundhenk et al. 2016).

Thus, because (1) STT to the PBL is so important during spring and (2) very little research has documented STT and TME forecast skill during spring, we feel that focusing on spring rather than other months is a worthwhile endeavor. That said, we agree with the reviewer that we should be more explicit when explaining our motivation for examining spring, so we have rewritten a portion of the Introduction and added some additional that expands on our rationale for focusing on MAM (see lines 31-46).

Reviewer wrote: *Lines 185-187: It is not clear how the claim of "vertically deep" shifts in the jet stream is made from the 2-D comparison. I am assuming that is because the jet climatology is derived from vertically averaged winds between 100-500 hPa. However, given the strong vertical gradients of wind around the jet stream, the wind changes don't have to be vertically deep to result in shifts to the jet. In addition, the phrase could also be interpreted to imply the jet stream location shifts vertically which is not necessarily the case.*

Our response: You are correct in assuming that the jet climatology was calculated for vertically averaged winds between 100-500 hPa. Although we did not include the results in the manuscript, we also tested our results using a "shallow jet" climatology (based on upper tropospheric winds only, which was also taken from the Koch et al. 2006 dataset). We found that our results were not dependent on the jet climatology depth. Thus, it would seem that the depth is probably not super important, and given the potential for confusion that you have pointed out, we have simply removed the reference to jet depth.

Reviewer wrote: *Line 223 and Table 2: Please add the significance level information (e.g. what is >95% significance level?). The correlations are much less meaningful without this information.*

Our response: Thank you for pointing out this omission. We included p-values in Table 1 and confidence intervals in Fig. 4, but for some reason neglected to do so for Table 2. However, instead of listing p-values in Table 2, we have listed the 95[th] percentile confidence intervals (we include the p-values in the Table 2 caption). We made this choice

for two reasons. First, we have some concerns about whether the p-values are really meaningful for the results in Table 2. For example, the p-value for PC3 at week 6 is <0.05 despite the fact that the correlation is 0.09; that is, we feel that the 0.09 correlation is essentially useless for prediction purposes despite the fact that it is technically "statistically significant". A second reason to include confidence intervals instead is that they at least provide some amount of information about the "usefulness" of the correlation. Indeed, this is the reason that we included confidence intervals for the similar forecast-verification time series correlation calculations shown in Fig. 4.

Nevertheless, for completeness, we did note what the p-values are for the full time series in the Table 2 caption, but we also added what amounts to a note of caution to readers who might over-interpret the "significance" of the small p-values. Specifically, we included additional p-value information where we tried to take into account autocorrelation in the data that might artificially boost the p-values. This additional calculation shows that indeed the PC3 correlations at weeks 5 and 6 are likely of negligible usefulness/statistical significance.

Reviewer wrote: *There are several instances where the references cited only include recent papers and could benefit by being more historical. For examples: lines 201-202, previously cited work by Reed and Shapiro should be included here too; and Line 358, Olsen, et al., JGR, 2013.*

Our response: Good point. We have updated the citations to be more complete where you have mentioned; we also added the Olsen et al. reference on line 43 (newly added text).

Reviewer wrote: *Fig 10 and related discussion: There are multiple regions and 3 transport types considered. Figure 10 and the discussion concentrates on only 4 of these possible combinations. Are the results not statistically significant for the other area/transports that are not discussed? Or is the discussion representative of the other areas too? Some comment on this should be made in the paper.*

Our response: We did check many other combinations, but for the most part, the other combinations are either for regions nearby to what we are already showing ($STT_{500}$ and TME over the central Pacific) or are not statistically significant. We have added a sentence on lines to make a note of this on lines 355-358.

In addition to this new text, it is worth noting that on lines 412-415 we discuss the fact that TME over Alaska for positive phase EOF1 and negative phase EOF2 appears not to be reliably predictable, while on lines 372-375, we discuss the lack of significant forecast skill over the western US.

Reviewer wrote: *Many of the figures lack labels on the color bars and/or the axes. These need to be added.*

Our response: We find this comment a little confusing because (1) all of the figures have color bars and the units are noted in the figure captions, and (2) all of the figures have axes labels, we just chose to only include the axes labels at the bottom of every column and on the far right-hand side (or sometimes left-hand side) of each row. We prefer to keep this figure label convention as we have it because we believe that it helps to reduce the clutter on multi-panel plots (e.g., Figure 3 would have 18 sets of labels instead of 6, which we find to be preferable).

Reviewer wrote: *The color bar placements in Figure 1 should be better aligned to indicate which figures they are applicable to*

Our response: We have altered figure 1 so that each of the three columns has its own color bar.

Reviewer wrote: *What are the contour levels in Figure 2?*

Our response: Thank you for noticing this omission, we have added contour information to the figure caption.

Reviewer wrote: *Fig. 10: It would be greatly helpful if the color coding of the lines were shown in a legend. Also, why are the medians and whiskers not shown for the verifications? Please include them since it would be useful in the comparisons.*

Our response: Done.

Reviewer wrote: *Lines 138-142: Please add: "...jet variability [in Section 3.1], we compare the EOFs. . ." for clarity.*

Our response: Done.

Reviewer wrote: *Line 192: "loading" likely more appropriately "magnitude"*

Our response: Done.

---

## Author Comment (AC2) · 18 Feb 2021

*Response to Reviewer #2's Comments on:*
"***Subseasonal prediction of springtime Pacific-North American transport using upper-level wind forecasts***"
by J. Albers et al. (WCD paper # wcd-2020-60)

We thank Reviewer #2 for their thoughtful comments and suggestions, which greatly improved the science and readability of the manuscript. We address the Reviewer's concerns in detail below.

**Reviewer wrote:** *Not a concern, but a thank you for a very concise and very clear introduction! It is a little long, but provides a very introduction into the topic and reads very fluently.*

**Our response:** Thank you very much for the positive feedback. We were a little concerned about the length too, but decided that contextual clarity was important, so it's good to hear that this was helpful. That said, for the revision we did make an effort to try and shorten the Introduction up a little bit to improve readability.

**Reviewer wrote:** *The only aspect that serves a little more attention is the specific role that is attributed to tropopause folds and PV streamers, as proxies for Rossby wave breaking events. It would be nice if the text gets more specific (lee speculative) and possibly also incorporates some feature based analysis.*

> And then later in the review the reviewer wrote:

*Tropopause folds, Rossby wave breaking and PV streamers are mentioned at several places in the text. Their role, however, remains somewhat unclear. My specific concerns/questions in this context are: What does it mean that a PV streamer extends down to the surface? Are TMEs are really mentioned in Shapiro and Keyser (1990) and Koch et al. 2006) to be linked to Rossby waveguides? Furthermore, the feature-based dataset presented in Sprenger et al. (2017) also includes tropopause folds and PV streamers (as proxies for Rossby wave breaking); would it, therefore, be worthwhile to support the statements in this section by the corresponding composites of these features, as they are shown for TME and STT. I think that the physical argument given by the authors is essentially correct, but it would be nice to support it by the corresponding composites, if they can 'easily' be calculated.*

**Our response**: We definitely understand the reviewer's perspective regarding including attribution to folds versus streamers (and PV cutoffs should probably be included in this discussion as well). Indeed during initial exploration of our results, we did spend a fair bit of time creating and considering feature-based composites of folds, streamers, and PV cutoffs. In the end we decided not to include them because we already had 17 figures in the full manuscript (eleven figures in the main manuscript, plus one in the Appendix, plus five more Supplemental figures), which we judged to be on the edge of too many.

However, prompted by your suggestion, we have decided to include a few sentences (lines 272-273 and 279-296) that note (in very general terms) the relative roles of folds, streamers, and PV cutoffs in the context of the retrospective analysis (the description of the new fold/streamer/cutoff datasets is included on lines 227-235). To support these new sentences, we have also included several new fold, streamer, and cutoff composites in the Supplement (Figs. S5-S9). Beyond the sentences that are included, we are a little apprehensive about including too detailed of a discussion because (1) it is not the main subject of the manuscript (we are really interested in predicting the STT and TME, rather than the proximate causes of that transport), and (2) definitively linking (in a scientifically defensible manner) the STT and TME to folds, streamers, and cutoffs would probably require a whole manuscript itself, and we are a little uncomfortable speculating without having done a careful analysis.

In addition to the inclusion of the new composites and text just mentioned, we have also tried to address your specific comments from this portion of your review (see next three responses immediately below).

**Reviewer wrote:** *L187-190:*
> *"While the EOF patterns likely combine jet variability due to both the subtropical and polar front jets (Koch et al. 2006), a strong jet stream of either type will act as a waveguide for Rossby waves (e.g., Schwierz et al. 2004; Rivière 2010 and references therein) with an increased frequency of STT and TME (e.g., Shapiro and Keyser 1990, Koch et al. 2006)."*

[and later]

*Are TMEs are really mentioned in Shapiro and Keyer (1990) and Koch et al. 2006) to be linked to Rossby waveguides?*

**Our response**: You are right that we should be a little more careful with the reference structure here. That is, the Shapiro/Keyser paper is really only related to STT (not TME), while the Koch et al. paper is only related to STT and TME in the sense that STT and TME are related to Rossby waves that tend to be "steered" along the waveguide of the jet. We have switched out the Koch et al. reference for Sprenger et al. 2017 where they discuss the connection between TME, STE, and jet variability, Eady growth rates, etc. We have also added the Higgins et al. 2000 reference to help readers explore the connection between Pacific jet (and teleconnection) variability and TME.

**Reviewer wrote**: *L200-202:*

*"STTPBL on the other hand (Fig. 3 middle row), have maxima slightly downstream of the 500 hPa maxima, which reflects the fact that deep STT tends to occur as maturing Rossby waves amplify and tropopause folds and potential vorticity streamers extend downwards towards the surface (Wernli and Bourqui 2002; Sprenger et al. 2003, Appenzeller et a. 1996, Wernli and Sprenger 2007, Škerlak et al. 2015)."*

[and later]

*What does it mean that a PV streamer extends down to the surface?*

**Our response:** Thank you for pointing out that this section needs to be more carefully written. The point that we were trying to make (perhaps unclearly), was that as a Rossby wave breaks, filamented structures are stretched out along isentropes. And because isentropes slope downwards towards the equator, any filaments that are stretched out towards the equator will consequently "extend downwards" closer to the surface of the Earth. This view is consistent with Škerlak et al. 2014 (see first paragraph of their Sect. 3.1.3 and their Fig. 4), where they point out that local maxima in STT tend to move equatorward as pressure increases (i.e., maxima in STT are farther poleward at 500 hPa than the maxima at 800 hPa).

We have tried to rewrite the text in this section (lines 275-280) to be more careful with our language (note that this section now also includes the mention of the streamer and PV cutoff figures discussed above). One issue here is that it is a little unclear exactly how the mass is exchanged to the PBL, because it is unclear from the time averaged STT climatologies how the isentropic surfaces are warped as the Rossby wave fully breaks (obviously in a climatological sense, the isentropes that intersect the stratosphere in midlatitudes do not intersect the surface over western N. America for example.) In that sense, a better word choice than "extends to the surface" is to say that streamers "extend towards the surface", because clearly at some point turbulent mixing and diabatic circulations must occur to cause irreversible mixing across isentropes. Note that our view is supported by discussions in Škerlak et al. 2014 (see the left column on their page 918), which we now reference on lines 277-278.

**Reviewer wrote:** *Whereas the signals for STT-500hPa and TME are rather clear, it is much more difficult to see the signal of STT-PBL. For instance, in Figure 7 it is really difficult to see the signal in the west-American box for the hindcast simulations. The same applies, to a lesser degree, to the retrospective analysis in Figure 3. I wonder whether this would become somewhat clearer if (especially in Figure 7) the colorbar is adjusted. Overall, I have have the impression the link between the jet structures, as expressed in PC1-3, is clearly discernible in for STT-500hPA and TME, but rather weak for STT-PBL. Actually, Figure 10 shows that there is some skill, but it is difficult to get it from the Figure 7.In summary, I see that the authors are aware of this fact and argue by means of Rossby waves not extending to the surface (see point 2 above) and PBL effects (altitude) to explain this weak signal of STT-PBL compared to STT-500hPA and TME, but the explanation is not fully convincing. Please add, at least, some references supporting the argument..*

**Our response**: We totally agree that the STT-PBL is somewhat hard to see relative to STT-500 and TME! Rather than a flaw in the choice of color scale though, we see this as fundamentally indicative of the fact that the STT-PBL anomalies are rather weak and difficult to predict. Nevertheless, to help the reviewer confirm that the Fig. 7 patterns

of STT-PBL are consistent (spatially) with those shown in Figs. 3 and S3, we include a figure below that shows STT-PBL with a modified color bar and a slightly different color scheme that is easier to see (the figure below is analogous to Fig. 7 in the main manuscript). Still, in the published version of Fig. 7, we have decided to keep the scale of the color bar for STT-PBL equivalent to that used for STT-500 because we feel that changing the color scale to make the STT-PBL anomalies easier to see risks making them seem larger or more robust than they actually are (that is, when a reader thinks that the anomalies are hard to see, thus concluding that the result/signal is weak, we feel that they are correctly interpreting the reality of the problem).

To address the reviewer's concern that we did not fully explain the reason that the forecasted STT-PBL signal is weak compared to STT-500, we have expanded the text (see lines 385-409) to include a discussion highlighting potential reasons why the STT-PBL forecasts might be worse than those for STT-500.

[Figure]

**Reviewer wrote**: *L48: As a more recent study linking STT and PV streamers, the authors might want to add the following reference: Sprenger, M., Wernli, H., & Bourqui, M. (2007).*

**Our response:** Done.

**Reviewer wrote**: *L51 "extratropical transport related to local frontal dynamics" This is a little too narrow, also given the references given in brackets; the references actually refer to warm conveyor belts (WCB) and, therefore, it might be appropriate to bring up this term and explain also in one, two sentences how WCBs, TMEs and atmospheric rivers are related. A possible reference could be:*

**Our response**: This is a very helpful comment, particularly because we (the authors) are not atmospheric river/TME specialists. We have updated the text to state only the most important physical processes involved with intense water vapor transport events (i.e., ARs, WCBs, and TMEs). In short, we decided to keep the text short here because we are concerned that discussing the physics (and differences) of ARs, WCBs, and TMEs would not be possible with just a few sentences and would end up being distracting to the flow of the Introduction (which as you pointed out earlier in this review, is already a bit lengthy). Instead, we have added a few extra citations that should help readers interested in understanding the differences (and similarities) between ARs, WCBs, and TMEs (additional citations now included are: Sodemann et al. 2020, Knippertz and Martin 2007, and Ralph et al. 2018).

**Reviewer wrote**: *L60: "deep STT is limited" -> "deep STT into the PBL is limited"; to make very clear that 'deep' refers to PBL-reaching trajectories*

**Our response**: Good suggestion, fixed.

**Reviewer wrote**: *L103: The retrospective (1979-206) and hindcast (1997-2016) analysis periods are introduced here, which is fine. But at first reading, I wondered a little why these time periods are chosen and what exactly are the underlying datasets. This becomes very clear only in section 2.1, but would have 'helped' me already at this place.*

**Our response**: Interestingly, a co-author made the same suggestion. To address this issue, several parenthetical comments and an additional reference have been inserted that help highlight the datasets used and the time periods that they cover (see lines 137-142).

**Reviewer wrote**: *Figure 1: I am somewhat confused how I have to link the two color bars to the different panels. This should be mentioned, at least, in the figure caption. Furthermore, I wonder whether the unit 'events/6-hourly time step' could be replaced by an equivalent, but more intuitive 'frequency per month'?*

**Our response**: We have altered the figure so that each of the three columns has its own color bar. Regarding the units, to be completely honest, we don't find the units to be terribly intuitive either, but those are the units that are defined in the original dataset. This confusion is one of the reasons that we converted the anomalies to units of standard deviations, which we found to be easier to grasp (clearly that doesn't work for plotting a climatology). We are not so sure that converting the units to events/month would help, so we prefer to just leave them as is. Luckily the purpose of the climatology figure is just to give readers a broad idea of the geography and seasonal timing of STT and TME, so the exact numbers are not so critical.

**Reviewer wrote**: *- Figure 2: The jet variability is expressed by the first three components of an EOF analysis. This is fine, and the three EOF patterns in Figure 2 clearly show distinct jet signatures (structures). Just to be sure: Would it be correct to name EOF1 a 'double-jet structure', EOF a 'single jet structure and EOF3 a 'transitional structure'? The patterns are OK, but it would be nice to attribute them to a somewhat less abstract (EOF loadings) picture.*

**Our response**: Unfortunately, probably not. It is perhaps best to simply interpret the wind EOF patterns as regions where jet variance increases or decreases, which in essence just amounts to regions where the jet spends more/less time, and consequently, wave breaking is more/less frequent. In general, each of the patterns represents some amount of jet retraction/extension and/or north/south shifts in the location of the jet axis. You can see that in the figures we have included immediately below where the EOF wind patterns were either added to, or subtracted from, the wind climatology.

Climatology plus EOF patterns:

[Figure]

Climatology minus EOF patterns:

[Figure]

**Reviewer wrote**: *L136: "is very little amplitude remaining"; What does 'amplitude' mean in this context? Would 'skill' be the better word?*

**Our response**: While it is true that there is less skill at the longer leads (correlation-wise), what we are referring to here is actually the amplitude of the anomaly that is forecasted. To understand why we discuss the anomaly amplitude, consider our Fig. 4 as an example. When Fig. 4 is recreated for week 1 instead of weeks 3-5, the amplitude of the orange lines matches the verification black line almost identically. But for weeks 3-5 as shown in Fig. 4, you will notice that the orange line forecasted time series is consistently of smaller anomaly amplitude than the verification. This is a common feature of forecasts at increasingly longer lead times. To make this more clear, we have modified the text of this whole section (see lines 176-179).

**Reviewer wrote**: *L152-153: "units of number of mass exchange events per 6-hourly time step"; It might be worthwhile in one sentence more clearly what this unit means. How is a 'mass exchange event' defined?*

**Our response**: Explaining how a mass exchange event is defined is rather involved, and since it is not our algorithm, we think that it is best that readers go to the original sources (which we cite in the text) for information about these calculations.

**Reviewer wrote**: *L218: "in terms of both correlation and amplitude"; really 'correlation', or should it be 'location'?*

**Our response**: We mean correlation and anomaly amplitude (see discussion about forecast anomaly amplitude in our previous response above). To be more clear in the text, we have added the word 'anomaly' before 'amplitude'.